# TR-BEACON: Shedding Light on Efficient Behavior Discovery in High-Dimensional Spaces with Bayesian Novelty Search over Trust Regions

**Wei-Ting Tang**
The Ohio State University
tang.1856@osu.edu

**Ankush Chakrabarty**
Mitsubishi Electric Research Laboratories
achakrabarty@ieee.org

**Joel A. Paulson**
The Ohio State University
paulson.82@osu.edu

## Abstract

Novelty search (NS) algorithms automatically discover diverse system behaviors through simulations or experiments, often treating the system as a black box due to unknown input-output relationships. Previously, we introduced BEACON, a sample-efficient NS algorithm that uses probabilistic surrogate models to select inputs likely to produce novel behaviors. In this paper, we present TR-BEACON, a high-dimensional extension of BEACON that mitigates the curse of dimensionality by constructing local probabilistic models over a trust region whose geometry is adapted as information is gathered. Through numerical experiments, we demonstrate that TR-BEACON significantly outperforms state-of-the-art NS methods on high-dimensional problems, including a challenging robot maze navigation task.

## 1 Introduction

Novelty search (NS) refers to a class of methods that, as opposed to minimizing or maximizing over specific objectives, aim to search for diverse system outcomes [1, 2]. Mathematically, we assume our system outcomes are determined by evaluating a function $\boldsymbol{f} : \mathbb{R}^D \to \mathbb{R}^M$ that maps a $D$-dimensional input vector $\boldsymbol{x} \in \mathbb{R}^D$ to a $M$-dimensional outcome space. The inputs typically represent design parameters that we can tweak in our system and the outcomes correspond to important dimensions of variation. In airplane design, $\boldsymbol{x}$ would represent things like the size and shape of the wing while $\boldsymbol{f}(\boldsymbol{x})$ represents the collection of all important measurable quantities like the drag coefficient, life, structural integrity, and fuel efficiency. For drug molecules, $\boldsymbol{x}$ could represent different molecular substitution patterns while $\boldsymbol{f}(\boldsymbol{x})$ represents the size, effectiveness, and stability of the molecules. NS has demonstrated strong potential for solving so-called *deceptive* optimization problems where it is common for many existing optimization algorithms get stuck in local optima [3, 4, 5].

Assuming we have evaluated $\boldsymbol{f}$ at $n > 0$ input values, yielding dataset $\mathcal{D}_n = \{(\boldsymbol{x}_i, \boldsymbol{f}(\boldsymbol{x}_i))\}_{i=1}^n$, a popular way to measure the "novelty" is the average distance to the $k$-nearest neighbors as follows

$$\rho(\boldsymbol{x}|\mathcal{D}) = \tfrac{1}{k} \sum_{i=1}^{k} \text{dist}(\boldsymbol{f}(\boldsymbol{x}), \boldsymbol{y}_i^\star), \tag{1}$$

where $\{\boldsymbol{y}_1^\star, \ldots, \boldsymbol{y}_k^\star\}$ are the $k$ closest outcomes to $\boldsymbol{f}(\boldsymbol{x})$ and $\text{dist}(\cdot)$ is any valid distance metric in $\mathbb{R}^M$. If we had knowledge of $\boldsymbol{f}$, we could sequentially select inputs that maximize novelty at every iteration, i.e., $\boldsymbol{x}_{n+1} = \text{argmax}_{\boldsymbol{x} \in \mathcal{X}} \rho(\boldsymbol{x}|\mathcal{D}_n)$ over our design space $\mathcal{X} \subset \mathbb{R}^D$. The key challenge in

Workshop on Bayesian Decision-making and Uncertainty, 38th Conference on Neural Information Processing Systems (NeurIPS 2024).

practice is that $\boldsymbol{f}$ is often a black-box function such that we cannot solve this optimization problem exactly. To overcome this challenge, the vast majority of existing NS methods rely on meta-heuristics, such as evolutionary algorithms, to select new evaluation points, e.g., [1, 6]. These methods can be effective when we have a large evaluation budget (e.g., can evaluate $\boldsymbol{f}$ several thousand or more times); however, for many applications of interest, we have expensive evaluations. For example, in the airplane design problem, evaluating $\boldsymbol{f}(\boldsymbol{x})$ may require a complex computational fluid dynamic simulation or physical experiment. These problems naturally require a much smaller evaluation budget for which existing NS methods are not designed.

Recently, the authors proposed an NS method, called BEACON, specifically designed for expensive black-box functions [7]. BEACON constructs a multi-output Gaussian process surrogate model for $\boldsymbol{f}$ that is combined with a novel acquisition function for trading off exploration and exploitation of the input space, which enables efficient sampling of the search space. However, BEACON relies on a global surrogate model and thus is inherently limited by the curse of dimensionality. Although we previously explored the use of custom priors and kernels to deal with this challenge, these require $\boldsymbol{f}$ to exhibit special structure and are likely to fail when these assumptions do not hold.

Motivated by this challenge, we propose a *local* variant of BEACON for high-dimensional NS (or behavior discovery) problems. Our proposed approach, TR-BEACON, leverages the notion of a trust region (TR) [8] to limit the size of the search space over which we need to build our surrogate model. Trust regions are commonly applied in the optimization literature and have been used within the TuRBO algorithm [9] for high-dimensional Bayesian optimization. A key component of TR-BEACON is the development of a new trust region management system that at every iteration (i) computes a center point based on the most novel data point observed so far and (ii) uses a variance-based measure to decide if the trust region should be enlarged or shrunk. The next section provides a detailed description of TR-BEACON, which is followed by numerical experiments on a synthetic objective and a challenging robot maze navigation problem from the NS literature that demonstrate its strong empirical performance on high-dimensional problems.

## 2 TR-BEACON: Algorithm Description

A complete description of TR-BEACON is given in Algorithm 1. The first key component is the construction of a (local) multi-output Gaussian processes (MOGPs) [10] to model the $M$-dimensional outcome space. Recall that a GP is a non-parametric model that learns a probabilistic surrogate function from (potentially) noisy data. To build an MOGP, we introduce an additional input variable $j \in \{1, \cdots, M\}$ to describe the $j^{\text{th}}$ element of the outcome vector $[\boldsymbol{f}(\boldsymbol{x})]_j$. Given observed data $\mathcal{D}_n$, the expressions for the posterior mean and variance of the MOGP can be derived analytically; see Appendix A. The second key component is the choice of the "trust region" (TR). Here, we choose the TR to be a hyperrectangle with center equal to the input that gave the output that is furthest away from its neighbors, denoted by $\boldsymbol{x}_n^c$. At the start of the method, we set the base side length of the TR to $L \leftarrow L_{\text{init}}$. Following TuRBO, the base side length is rescaled according to the estimated average lengthscale $\ell_i$ for dimension $i$ across the different outputs in the MOGP model such that $L_i = \ell_i L / (\prod_{j=1}^D \ell_j)^{1/D}$. As commonly done in TR methods, if the optimizer "makes progress", the TR will be expanded whereas, if it "fails", the TR will be shrunk. We double and halve the size of the TR after, respectively, $\tau_{\text{succ}}$ consecutive successes and $\tau_{\text{fail}}$ consecutive failures. In addition, we do not let the TR length exceed a size $L_{\max}$ and reset the length to $L_{\text{init}}$ if $L$ falls below a minimum threshold $L_{\min}$. Since the novelty measure (1) changes at every iteration (depends on the full set of collected data), we cannot define a "success" as simply improving upon the best-known value. Instead, we define a "success" as choosing a candidate that increases the total variance of the outcomes and a "failure" as choosing one that does not. The last critical component is the choice of acquisition function inside of the TR for which we use with the same approach as BEACON, i.e.,

$$\boldsymbol{x}_{n+1} \in \underset{\boldsymbol{x} \in \Delta_n}{\operatorname{argmax}} \ \tfrac{1}{k} \mathbf{e}_k^\top \operatorname{sort}\left(\left[\operatorname{dist}(\boldsymbol{g}(\boldsymbol{x}), \boldsymbol{y}_1), \ldots, \operatorname{dist}(\boldsymbol{g}(\boldsymbol{x}), \boldsymbol{y}_n)\right]^\top\right), \tag{2}$$

where $\boldsymbol{g} \sim \boldsymbol{f} | \mathcal{D}_n$ is a Thompson sample (TS) from the MOGP posterior, $\Delta_n$ denotes the TR at the current iteration $n$, $\mathbf{e}_k$ is a vector of ones in the first $k$ entries and zeros in the remaining entries, and sort($\cdot$) is the standard sort operator that sorts its input in descending order. The formulation can be thought of as a differentiable version of (1) where the true unknown function $\boldsymbol{f}$ is replaced with a TS $\boldsymbol{g}$. We let $\mathcal{D}^{\boldsymbol{x}}$ and $\mathcal{D}^{\boldsymbol{y}}$ denote the subset of input and output values in the dataset $\mathcal{D}$, respectively. An illustration of the evolution of TR-BEACON is shown in Fig. 1 for a simple 2-D function. We see

**Algorithm 1** TR-BEACON

---

1: **Input:** Initial data $\mathcal{D}$, input domain $\mathcal{X}$, outcome function $\boldsymbol{f}$, outcome distance metric $\text{dist}(\cdot)$, MOGP prior $\mathcal{MOGP}(\boldsymbol{\mu}, \boldsymbol{\kappa})$, and hyperparameters $L_{\text{init}}, L_{\text{max}}, L_{\text{min}}, \tau_{\text{succ}}, \tau_{\text{fail}}$, and $k$.

2: **Initialize:** Data $\mathcal{D}_0 \leftarrow \mathcal{D}$, success metric $\tau_0 \leftarrow \text{trace}(\text{Cov}(\{\boldsymbol{y}\}_{\boldsymbol{y} \in \mathcal{D}_0^{\boldsymbol{y}}}))$, trust region base length $L^{(0)} \leftarrow L_{\text{init}}$, and success/failure counters $n_{\text{succ}} = n_{\text{fail}} = 0$.

3: **for** $n = 0, 1, 2, \ldots,$ **do**

4:   Train MOGP surrogate model $\mathcal{MOGP}_n$ given all available data $\mathcal{D}_n$.

5:   Set the trust region center as $\boldsymbol{x}_n^c = \boldsymbol{x}_{i^\star}$ where $i^\star = \text{argmax}_{i=1,\ldots,q} \sum_{j \neq i} \text{dist}(\boldsymbol{y}_i, \boldsymbol{y}_j)$.

6:   Set trust region lengths as $L_i \leftarrow \ell_i L / (\prod_{j=1}^{D} \ell_j)^{1/D}$ given average lengthscales $\{\ell_i\}_{i=1}^{D}$.

7:   Construct the trust region set $\Delta_n \leftarrow \boldsymbol{x}_n^c \oplus [-0.5L_1, 0.5L_1] \times \cdots \times [-0.5L_D, 0.5L_D]$.

8:   Get next sample point $\boldsymbol{x}_{n+1}$ by maximizing the acquisition function, as shown in (2).

9:   Evaluate expensive outcome function at new sample point to get $\boldsymbol{y}_{n+1} \leftarrow \boldsymbol{f}(\boldsymbol{x}_{n+1})$.

10:   Update data $\mathcal{D}_{n+1} \leftarrow \mathcal{D}_n \cup \{(\boldsymbol{x}_{n+1}, \boldsymbol{y}_{n+1})\}$ and calculate metric $\tau_{n+1} \leftarrow \text{trace}(\text{Cov}(\{\boldsymbol{y}\}_{\boldsymbol{y} \in \mathcal{D}_{n+1}^{\boldsymbol{y}}}))$.

11:   **if** $\tau_{n+1} > \tau_n$ **then**

12:    $n_{\text{succ}} = n_{\text{succ}} + 1$ and $n_{\text{fail}} = 0$.

13:   **else**

14:    $n_{\text{succ}} = 0$ and $n_{\text{fail}} = n_{\text{fail}} + 1$.

15:   **end if**

16:   **if** $n_{\text{succ}} == \tau_{\text{succ}}$ **then**

17:    Enlarge the trust region $L^{(n+1)} \leftarrow \min\{2L^{(n)}, L_{\text{max}}\}$ and reset counter $n_{\text{succ}} = 0$.

18:   **else if** $n_{\text{fail}} == \tau_{\text{fail}}$ **then**

19:    Shrink the trust region $L^{(n+1)} \leftarrow L^{(n)}/2$ and reset counter $n_{\text{fail}} = 0$.

20:   **end if**

21:   **if** $L^{(n+1)} < L_{\text{min}}$ **then**

22:    The trust region is below the minimum size; stop the optimization process.

23:   **end if**

24: **end for**

---

that our proposed approach is able to efficiently explore the space of outcomes (finding many values along the edge of the support of the distribution of outcomes), even though the MOGP model does not accurately capture the novelty metric surface especially in the early iterations.

## 3 Numerical Experiments

We compare TR-BEACON with several state-of-the-art (SOTA) baseline algorithms on a synthetic and a reinforcement learning (RL) problem. Details for all methods are included in Appendix B. All experiments can be reproduced with the code provided in https://github.com/PaulsonLab/TR-BEACON.git.

### 3.1 20-D Ackley

For the first case study, we consider a 20-D Ackley synthetic function; see Appendix C for its analytical form. We consider the domain $x \in [-2, 2]$ and generate 40 initial samples using the Sobol sequence [11] to train a GP with a standard radial basis function (RBF) kernel. We use reachability as the performance metric, which corresponds to the number of bins visited by the algorithms given a finite number of evaluations (50 equally sized bins are chosen, with each bin effectively representing a behavior we would like to discover). Results are shown in Fig. 2a over 10 replicates of randomly selected initial points (the dark lines represent the mean and the shaded regions correspond to 95% confidence regions). We see that TR-BEACON achieves the highest reachability of all the methods, with BEACON performing second best as the performance drops off due to the difficulty of globally modeling the outcome over a 20-D input space.

### 3.2 24-D Reinforcement Learning Problem - Maze Navigation

Next, we examine a challenging RL problem from the OpenAI Gym [12] environment. The task is to move a ball from its starting location to a target location within a given number of time steps (see Appendix D for details). We define the actor policy using a bias-free feed-forward neural network. We want to compute the optimal set of weight parameters that can maximize the reward of successfully

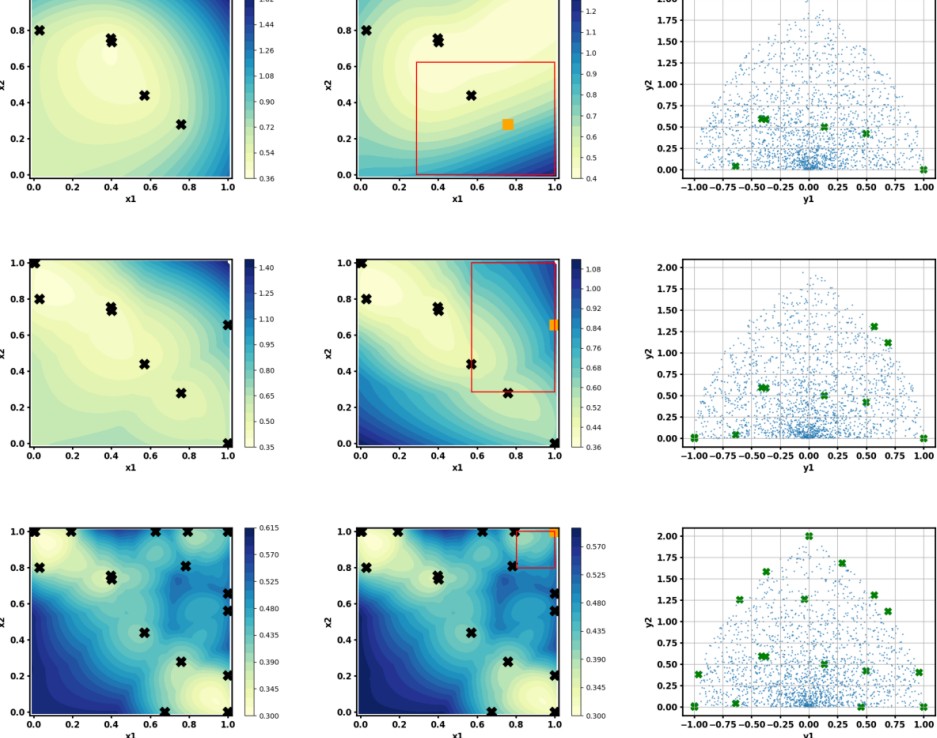

Figure 1: Illustration of TR-BEACON on a 2-D synthetic function with 2 outcomes. As the algorithm proceeds (from top to bottom), TR-BEACON identifies novel outcomes from the unexplored local region and gradually shrink the TR accordingly. (*Left*) The contour for the true novelty metric by replacing $g$ in (2) with the true function. (*Middle*) The contour for the predicted novelty metric by replacing $g$ in (2) with the MOGP posterior mean. (*Right*) the scatter plot for sampled outcomes and the true function outcomes. (■: center point of TR; ✕: sampled features; ✕: sampled outcomes)

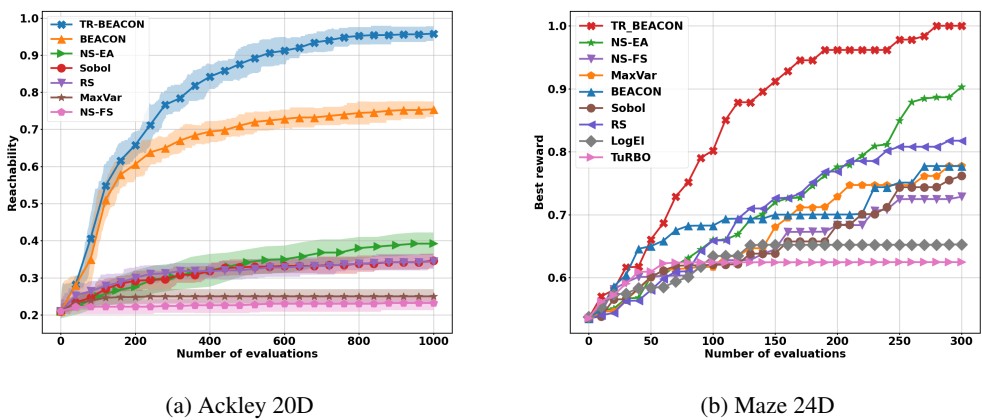

(a) Ackley 20D
(b) Maze 24D

Figure 2: Results on (a) 20-D Ackley function and (b) 24-D RL maze navigation problem

getting the ball to the exit. This is known to be a deceptive problem in the sense that the algorithm can often unexpectedly push the system to have the ball get trapped on a wall when directly optimizing the objective (distance to the exit) [13]. NS methods can overcome this challenge by considering the $(x, y)$ coordinates of the landing position as outcomes and biasing the samples toward discovering new outcomes. Fig. 2b shows the mean of the best reward found by all baseline methods across 30 replicates. The percentage of reaching the maximum achievable reward for all baseline methods is

shown in Table 1 in Appendix D. TR-BEACON substantially outperforms all considered baselines; it can in fact reach the maximum achievable reward within 300 episodes in all replicates. NS-EA has the second-best performance; however, only reaches $\sim 90\%$ of the best possible reward, with even lower performance in the earlier iterations. Lastly, two SOTA black-box optimizers, mainly TuRBO [9] and LogEI [14], interestingly exhibit the worst performance out of all tested methods, which is a consequence of the deceptive characteristics of this problem.

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

# A GPs and MOGPs

For GP, we assume that function outputs $y(x) = f(x) + \epsilon$, where $\epsilon \sim \mathcal{N}(0, \sigma^2 \boldsymbol{I}_n)$, are jointly Gaussian distributed for any finite collection of input $x$ [15]. For MOGP, the prior $h \sim GP(\mu, \kappa)$ is placed on the extended feature space $\mathbb{X} \times \mathbb{N}_{n_f}$. Any finite collection from the multi-output GP prior $h \sim \mathcal{GP}(\mu, \kappa)$ remains a GP, so we can write down the posterior $p(h|\mathcal{A}) \sim \mathcal{GP}(\mu_\mathcal{A}, \kappa_\mathcal{A})$ with the posterior mean and variance derived based on conditional Gaussian [15]:

$$\mu_\mathcal{A}(\boldsymbol{x}, j) = \mu(\boldsymbol{x}, j) + \boldsymbol{\kappa}_\mathcal{A}^\top(\boldsymbol{x}, j) \left( \boldsymbol{K}_\mathcal{A} + \sigma^2 \mathbf{I}_N \right)^{-1} \left( \boldsymbol{y} - \boldsymbol{\mu}_\mathcal{A} \right), \tag{3a}$$

$$\kappa_\mathcal{A}((\boldsymbol{x}, j), (\boldsymbol{x}', j')) = \kappa((\boldsymbol{x}, j), (\boldsymbol{x}', j')) - \boldsymbol{\kappa}_\mathcal{A}^\top(\boldsymbol{x}, j) \left( \boldsymbol{K}_\mathcal{A} + \sigma^2 \mathbf{I} \right)^{-1} \boldsymbol{\kappa}_\mathcal{A}(\boldsymbol{x}', j'), \tag{3b}$$

Where $\boldsymbol{\kappa}_\mathcal{A}(\boldsymbol{x}, j) \in \mathbb{R}^N$ is the covariance vector between the test data and the observed data, $\boldsymbol{K}_\mathcal{A} \in \mathbb{R}^{N \times N}$ is the covariance matrix of the observed data, $\boldsymbol{y} = [y_1, \ldots, y_N]^\top$ is the vector of observed outcomes, and $\boldsymbol{\mu}_\mathcal{A} = [\mu(x_1), \ldots, \mu(x_N)]^\top$ is the prior mean vector.

# B Method details

We provide the details for all methods considered in this work in this section.

## B.1 Hyperparameter settings for TR-BEACON

We use the following default values for all the hyperparameters of TR-BEACON: $L_{\text{init}} = 0.8$, $L_{\text{min}} = 0$, $L_{\text{max}} = 1.6$, $\tau_{\text{succ}} = 10$, $\tau_{\text{fail}} = D$, and $k = 10$.

## B.2 MaxVar

MaxVar [16] is a special setting for BO, where the acquisition function is defined as the trace of the MOGP posterior variance. The acquisition function can be expressed as follow:

$$\alpha_{\text{MaxVar}}(x) = \text{tr}(\Sigma_\mathcal{A}(x)) \tag{4}$$

MaxVar sequentially query new candidate by maximizing the above acquisition using L-BFGS-B algorithm in this work.

## B.3 LogEI

LogEI is a BO algorithm in which the acquisition function is defined as the logarithm of the expected improvement (EI). The Log EI acquisition function has been shown to effectively mitigate the gradient vanishing issue commonly encountered when optimizing the conventional EI acquisition function using gradient-based solvers [14]. The log EI acquisition function is defined as follow:

$$\log \text{EI}(x) = \log \left[ (\mu(x) - f^*)\Phi \left( \frac{\mu(x) - f^*}{\sigma(x)} \right) + \sigma(x)\varphi \left( \frac{\mu(x) - f^*}{\sigma(x)} \right) \right] \tag{5}$$

Where $f^*$ is the best function value found so far and $\Phi$ and $\varphi$ are cumulative and probability density function for standard normal distribution. We maximize Log EI with the L-BFGS-B algorithm for each iteration in this work.

## B.4 TuRBO

TuRBO [9] is a local BO algorithm relying on building local GP and searching potential candidates across one or more TRs. For each iteration $t$, TuRBO selects a batch of q new candidates $x_i^{(t)}$ that have the optimum function value across n independent TRs, as shown in Eq.(6). Each $\text{TR}_\ell$ has an independent posterior realization generated from Thompson sampling.

$$\mathbf{x}_i^{(t)} \in \text{argmin}_\ell \text{argmin}_{\mathbf{x} \in \text{TR}_\ell} g_\ell^{(i)} \quad \text{where} \quad g_\ell^{(i)} \sim \mathcal{GP}_\ell^{(t)} \left( \mu_\ell(\mathbf{x}), k_\ell(\mathbf{x}, \mathbf{x}') \right) \tag{6}$$

Unlike TR-BEACON that utilizes the efficient Thompson sampling method to generate differentiable posterior function realization, TuRBO generate posterior sample at a discrete candidate set and select the one that has the minimum posterior function value. In other word, no gradient-based algorithm is implemented to solve for (6).

## B.5 NS-EA

NS-EA (Novelty Search based on Evolutionary Algorithm) is a vanilla NS algorithm that replaces the objective-oriented fitness function with novelty-based fitness function in the evolutionary algorithm. The fitness function is shown in (7).

$$\rho(\boldsymbol{x}|\mathcal{D}_{\text{obs}}) = \frac{1}{K} \sum_{k=1}^{K} \text{dist}\left(\boldsymbol{f}(\boldsymbol{x}), \boldsymbol{y}_k^{\text{nn}}\right), \tag{7}$$

where $\{\boldsymbol{y}_1^{\text{nn}}, ..., \boldsymbol{y}_K^{\text{nn}}\}$ denotes the $K$-nearest observed outcomes, given an outcome $\boldsymbol{f}(\boldsymbol{x})$, and $\text{dist}(\cdot, \cdot)$ is a distance metric in $\mathbb{R}^{n_f}$, e.g., the 2-norm. The code for implementing NS-EA can be found in https://github.com/alaflaquiere/simple-ns.

## B.6 NS-FS

NS-FS (Novelty Search in Feature Space) is a NS-based algorithm proposed by Tang et al. [7]. It is similar to the idea of vanilla NS algorithm, in which we calculate the distance w.r.t. the feature space instead of the outcome space. NS-FS sequentially sample new candidate by maximizing the following acquisition function for each iteration:

$$\alpha_{\text{NS-FS}}(\boldsymbol{x}) = \sum_{i=1}^{k} \text{dist}_{\mathcal{X}}(\boldsymbol{x}, \boldsymbol{x}_i^{\star}),$$

This acquisition function calculates the sum of distance from $\boldsymbol{x}$ to its k-nearest feature $\boldsymbol{x}_i^*$. It is worth noting that NS-FS doesn't require the need to build surrogate model. The acquisition function $\alpha_{\text{NS-FS}}$ can be maximized by gradient-based solver such as L-BFGS-B implemented in this work.

## B.7 RS

Random search (RS) is one of the most commonly used baseline algorithm for optimizing black-box functions. It works by sequentially sampling new candidate from a uniformly distributed set of continuous/discrete candidates.

## B.8 Sobol

Sobol is a sampling algorithm that sequentially sample new point from the quasi-random low-discrepancy sequences. Unlike purely random sampling, Sobol sequences are designed to cover the space more uniformly, reducing the occurrence of clusters and gaps. Sobol sequence is a common baseline algorithm implemented in optimization literature due to its straightforward implementation and the guarantee of dense sampling in the limit of an infinite budget.

# C Ackley function

The D dimensional Ackley function can be expressed as follow [17]:

$$f(\boldsymbol{x}) = -a \exp\left(-b\sqrt{\frac{1}{D}\sum_{i=1}^{D} x_i^2}\right) - \exp\left(\frac{1}{D}\sum_{i=1}^{D} \cos(cx_i)\right) + a + \exp(1),$$

where $a = -20$, $b = 0.2$, and $c = 2\pi$. This function has a global minimum $y^* = 0$ locating at $x^* = (0, ..., 0)$.

# D Maze problem

In Figure 3, the starting location is represented by the green ball, while the target location is represented by the red ball. This problem involves four state variables (coordinates and velocity in the x and y directions) and two actions (linear force in the x and y directions). The process terminates when the Euclidean distance between the achieved location and the target location is less or equal to

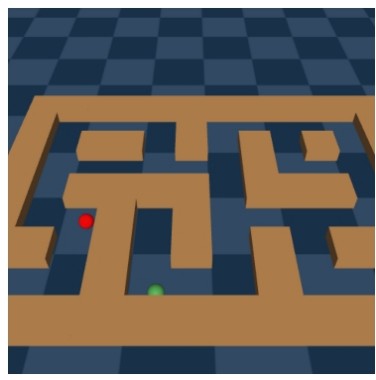

Figure 3: Environment for the maze navigation problem

Table 1: Percentage of reaching maximum possible reward for all tested methods across 30 replicates

| Algorithm | Best reward percentage |
|---|---|
| TR-BEACON | 100% |
| NS-EA | 77% |
| RS | 53% |
| BEACON | 47% |
| MaxVar | 43% |
| Sobol | 37% |
| NS-FS | 30% |
| logEI | 23% |
| TuRBO | 23% |

0.5, as per the OpenAI Gym default setting. We then define the reward function of this RL problem as follow:

$$\text{Reward} = \begin{cases} 1 & \text{if terminated,} \\ \frac{\text{initial distance from the target} - \text{final distance from the target}}{\text{initial distance from the target}} & \text{otherwise.} \end{cases} \quad (8)$$

We define the control policy as a bias-free single-hidden layer feedforward neural network with 24 weight parameters, structured as 4-4-2. We also include the percentage of reaching reward = 1 for all baseline methods, as shown in Table 1.

