# OpenReview forum: "TR-BEACON: Shedding Light on Efficient Behavior Discovery in High-Dimensional Spaces with Bayesian Novelty Search over Trust Regions"
_NeurIPS.cc/2024/Workshop/BDU — NeurIPS BDU Workshop 2024 Poster_

### Official Review · Reviewer_9k96 · 2024-09-18
**Review of TR-BEACON**

**Rating:** 6
**Confidence:** 3

**Review:**

Overall, the authors introduce TR-BEACON to extend their previously proposed BEACON to high-dimensional scenarios. The key idea is to leverage the trust region to limit search space size for building the surrogate model. Although trust region has been widely applied in diverse algorithms, the authors have developed a novel trust region management system and made a detailed algorithm description, which makes this work solid.

Strength:
[1] propose a novel trust region-based novelty search algorithm that can adapt to high-dimension scenarios.

[2] The authors give a detailed description of their proposed algorithm and spell out the difference between their early-stage works.

[3] The experimental results are extensive and convincing.

Weakness:

[1] Computation complexity analysis of the mentioned algorithms is lacking.

---

### Official Review · Reviewer_gK45 · 2024-09-26
**TR-BEACON: Shedding Light on Efficient Behavior Discovery in High-Dimensional Spaces with Bayesian Novelty Search over Trust Regions**

**Rating:** 7
**Confidence:** 3

**Review:**

1. Summary and contributions:

The main contribution of this paper is proposing a high-dimensional extension of BEACON. The authors implement local probabilistic models over trust regions, which effectively addresses the curse of dimensionality, allowing for efficient behavior discovery in complex systems.


2. Strengths and Weaknesses:

The paper is generally well-written and structured clearly. The authors clearly present the motivation of the paper and algorithms which makes me quite easy to read. Besides, experimental results demonstrate that TR-BEACON effectively identifies potential functional deficiencies. While the numerical experiments are convincing, they primarily focus on two specific tasks.  A broader range of applications could strengthen the argument for TR-BEACON's generalizability.

---

### Decision · Program_Chairs · 2024-10-09

Accept (Poster)